# Exploring white matter microstructure and the impact of antipsychotics in adolescent-onset psychosis

Claudia Barth[1]*, Vera Lonning[1], Tiril Pedersen Gurholt[1], Ole A. Andreassen[1], Anne M. Myhre[2], Ingrid Agartz[1,3,4]

1 NORMENT, Division of Mental Health and Addiction, Oslo University Hospital & Institute of Clinical Medicine, University of Oslo, Oslo, Norway, 2 Child & Adolescent Mental Health Research Unit, Oslo University Hospital, Oslo, Norway, 3 Department of Psychiatric Research, Diakonhjemmet Hospital, Oslo, Norway, 4 Centre for Psychiatry Research, Department of Clinical Neuroscience, Karolinska Institutet & Stockholm Health Care Services, Stockholm County Council, Stockholm, Sweden

* claudia.barth@medisin.uio.no

**Data Availability Statement:** All relevant data are within the paper and its Supporting Information files.

## Abstract

White matter abnormalities are well-established in adult patients with psychosis. Less is known about abnormalities in the rarely occurring adolescent early onset psychosis (EOP). In particular, whether antipsychotic medication might impact white matter microstructure is not known. Using 3T diffusion weighted imaging, we investigated differences in white matter microstructure and the impact of antipsychotic medication status in medicated (n = 11) and unmedicated (n = 11) EOP patients relative to healthy controls (n = 33), aged between 12–18 years. Using Tract-based Spatial Statistics, we calculate case-control differences in scalar diffusion measures, i.e. fractional anisotropy (FA), axial diffusion (AD) and radial diffusion (RD), and investigated their association with antipsychotic medication in patients. We found significantly lower FA in the left genu of the corpus callosum, the left anterior corona radiata (ACR) and the right superior longitudinal fasciculus in EOP patients relative to healthy controls. AD values were also lower in the left ACR, largely overlapping with the FA findings. Mean FA in the left ACR was significantly associated with antipsychotic medication status (Cohen's $d$ = 1.37, 95% CI [0.01, 2.68], $p$ = 0.008), showing higher FA values in medicated compared to unmedicated EOP patients. The present study is the first to link antipsychotic medication status to altered regional FA in the left ACR, a region hypothesized to contribute to the etiology of psychosis. Replications are warranted to draw firm conclusions about putatively enhancing effects of antipsychotic medication on white matter microstructure in adolescent-onset psychosis.

## Introduction

Psychotic disorders such as schizophrenia typically emerge during late adolescence or early adulthood, with debilitating consequences. Cases occurring earlier during adolescence are defined as early onset psychosis (EOP), with an age of onset before 18 years [1], and provide

**Funding:** This work was supported by the Research Council of Norway, grant numbers 223273, 213700, and 250358 (awarded to IA); the South-Eastern Norway Regional Health Authority, grant numbers 2016-118 and 2017-097 (awarded to IA); and Kristian Gerhard Jebsen Centre for Psychosis Research (SKGJ-MED-008, awarded to NORMENT). The funders had no role in study design, data collection and analysis, decision to publish, or preparation of the manuscript.

**Competing interests:** The authors have declared that no competing interests exist.

the exceptional opportunity to examine regional and disease-specific brain maturation after psychosis-onset.

EOP occurs rarely [2], affecting 0.05–0.5% of the population [1], and shows a heterogeneous clinical presentation. Relative to adult patients with psychosis, EOP patients show worse long-term prognoses [3], and EOP constitutes one of the leading causes of lifetime disease burden for adolescents aged 15–19 years [4]. While white matter microstructural alterations are well-established in adult onset psychosis [5], less is known about putative alterations in adolescent EOP. This is a critical research gap, as studying white matter abnormalities in adolescents with psychosis may provide important clues for understanding the pathophysiology of psychotic disorders.

Diffusion weighted imaging (DWI) is a magnetic resonance imaging (MRI) technique that maps the Brownian movement of water molecules in the brain *in vivo*. As axon membranes and myelin provide natural barriers for water diffusion, DWI can be used to infer local tissue properties [6]. Scalar measures of DWI include fractional anisotropy (FA), which characterizes the degree of diffusion directionality [6]. A decrease in FA has been linked to white matter damage [7]. Within each FA cluster, the relative contribution of axial diffusion (AD) along the primary axis, and radial diffusion (RD) perpendicular to it, can be investigated to specify diffusion directionality in more detail [6]. Both measures have been associated with different biological underpinnings. For instance, while increased RD has been associated with demyelination [8], decreased AD has been linked to axonal damage [6]. Scalar diffusion measures change across the lifespan. Throughout adolescence until early adulthood, FA increases and RD decreases in healthy individuals [9, 10], with males showing continuous white matter growth from childhood through early adulthood and females showing growth mainly during mid-adolescence [11]. The trajectories of AD are less known [9, 10].

In EOP patients, a number of studies show widespread FA reductions in several brain regions with low spatial overlap such as corpus callosum, cingulum, superior longitudinal fasciculus, inferior longitudinal fasciculus and fronto-occipital fasciculus, relative to healthy controls [12–17]. However, scalar DWI measures beyond FA are rarely analyzed in EOP populations. One study reports on increased RD values, indicative of potential demyelinating processes underlying the observed white matter abnormalities [16].

The low degree of regional specificity of whiter matter changes reported across studies seems attributable to a number of factors including differences in image acquisition, different analysis approaches (e.g. ROI vs voxel-wise), small sample sizes, low prevalence of EOP and differing sample characteristics such as age of onset and duration of illness. Further, antipsychotic medication status might also affect the pattern of white matter microstructure in EOP.

Studies investigating white matter microstructure in EOP mainly focus on case-control differences, either reporting antipsychotic effects as secondary findings or using antipsychotic medication status as a covariate of no interest. So far, studies in EOP patients do not indicate an impact of either current [18–20] or cumulative antipsychotic exposure [12, 19, 21] on regional scalar DWI measures. The absence of an antipsychotic dosage effect in EOP patients could reflect small sample sizes and young patients with shorter medication histories. However, large-scale cohort studies in adult-onset schizophrenia also do not find an effect of medication dosage on white-matter microstructure [5, 22]. Hence, medication dosage might not be a driving factor for potential effects of antipsychotic medication on the brain. So far, studies in EOP did not include unmedicated patients, which limits the ability to disentangle medication-mediated from disease-related effects on brain structure. Thus, zeroing in on unmedicated relative to medicated EOP patients provides the opportunity to investigate the impact of antipsychotic medication on white matter microstructure early in disease progression.

Here, we use a thoroughly clinically characterized adolescent EOP sample to (i) investigate white matter microstructure in comparison to healthy controls, and (ii) explore the association between antipsychotic medication and white matter microstructure in medicated compared to currently unmedicated/antipsychotic-naïve EOP patients. Using Tract-Based Spatial Statistics (TBSS), we calculate FA and its scalar sub-measures, RD and AD, and investigate their association with antipsychotic medication and other clinical measures (e.g. Positive and Negative Syndrome Scale, etc.). We hypothesized that EOP patients show widespread reduced FA attended by increased RD and unchanged AD compared to healthy controls [16], mainly in the corpus callosum and superior/inferior longitudinal fasciculus [17]. As there are no established effects of antipsychotic medication on white matter microstructure in EOP patients, our post hoc analyses are exploratory by nature.

## Methods and materials

### Participants

The study sample was drawn from the ongoing longitudinal Youth-Thematic-Organized-Psychosis (Youth-TOP) research study, which is part of the Norwegian Centre for Mental Disorders Research (NORMENT; https://www.med.uio.no/norment/english/) in Oslo, Norway. EOP patients, aged between 12–18 years, were recruited from in- and outpatient clinics in the Oslo region. Healthy controls were randomly selected from the Norwegian National Registry in the same catchment areas. All participants and their respective parents/guardians provided written informed consent. The study was approved by the Regional Committee for Medical Research Ethics (REK-Sør) and the Norwegian Data Inspectorate and was conducted in accordance with the Declaration of Helsinki.

For study inclusion, participants were required to have an intelligence quotient (IQ) > 70, a good command of the Norwegian language, no previous moderate to severe head injuries, no diagnosis of substance-induced psychotic disorder, and no organic brain disease. IQ was measured by the Wechsler Abbreviated Scale of Intelligence. Diagnosis was established according to the Diagnostic and Statistical Manual of Mental Disorder- IV criteria using the Norwegian version of the Kiddie-Schedule for Affective Disorders and Schizophrenia for School Aged Children (6–18 years): Present and Lifetime Version (K-SADS-PL, [23]). The clinical characterization was conducted by trained psychologists or psychiatrists.

A total of 67 participants (27 patients/40 controls) satisfied the inclusion criteria and were scanned with the same diffusion weighted imaging sequence. The larger control group was included in the study to provide a closer estimate of the control variance in adolescent populations. All MRI scans were visually inspected by a trained neuroradiologist to rule out any pathological changes. Out of the initial sample, seven control participants and five patients were excluded due to (i) clinical/radiological reasons (five patients/ three controls), or (ii) strong motion artefacts in the diffusion imaging data (four controls), resulting in a final sample of 55 participants (22 patients/ 33 controls) being entered in the statistical analysis. Strong motion artifacts were identified by visual inspection and by calculating the average motion per volume higher than 2 standard deviation above mean per participants using the motion estimates from the eddy current correction (see section diffusion data analysis).

### Clinical measures

Presence and severity of psychopathological symptoms of EOP patients were assessed using the Positive and Negative Syndrome Scale (PANSS, [24]). Children Global Assessment Scale (CGAS, [25]) and Mood and Feelings Questionnaire (MFQ, long version, [26]) were evaluated in all participants to measure general functioning level and to screen for depressive symptoms,

respectively. Recreational drug use was assessed within the structured K-SADS interview and scored with 0 or 1 for absent or present. For EOP patients, current and lifetime cumulative use of medication was recorded and converted into chlorpromazine equivalents (CPZ), using formulas published elsewhere [27]. While 11 EOP patients were off any antipsychotic medication at scan, yielding a lack of current CPZ values, three patients had received pharmacological treatment prior to inclusion, resulting in a low cumulative CPZ dosage for this subgroup. Nine patients were antipsychotic-naïve.

## MRI data acquisition

MR images were acquired on a 3-Tesla General Electric Signa HDxt scanner equipped with an 8-channel head coil at the Oslo University Hospital, Norway. The diffusion imaging data were acquired using a 2D spin-echo whole-brain echo-planar imaging sequence with the following parameters: slice thickness = 2.5 mm, repetition time = 15 s, echo time = 85 ms, flip angle = 90˚, acquisition matrix = 96 x 96, in-plane resolution = 1.875 x 1.875 mm. A total of 32 volumes with different gradient directions (b = 1000 s/mm$^2$), including two b0-volumes with reversed phase-encode (blip up/down), were acquired.

## Diffusion data analysis

Diffusion data were analyzed with FSL version 5.0.9 using the FMRIB's software library (https://fsl.fmrib.ox.ac.uk/fsl/fslwiki). Before creating voxel wise maps of diffusion parameters, the following steps of the standard processing pipeline were used: (i) *topup* to correct for susceptibility-induced distortions [28], (ii) *eddy* current correction to correct for gradient-coil distortions and head motion [29], (iii) removal of non-brain tissue using the Brain Extraction Tool (*bet*) [30], and (iv) local fitting of the diffusion tensor at each voxel using *dtifit* (FMRIB's Diffusion Toolbox (FDT) [31]). *Dtifit* yielded in voxel wise participant-specific maps of FA, mean diffusion (MD), and axial diffusion (AD, derived from eigenvector λ1). Based on the outputted eigenvectors λ2 and λ3, radial diffusion (RD) was computed ((λ2+λ3)/2)). We did not include MD in further analyses due to its lack of specificity [6]. Next, voxel wise statistical analysis of the FA data was carried out using TBSS [32]. First, all FA images were nonlinearly aligned to the most representative FA image out of all images and transformed into 1x1x1 mm$^3$ MNI152 standard space by means of affine registration. Secondly, TBSS projects all participant's FA data onto a mean FA tract skeleton (threshold FA > 0.25), before applying voxel wise cross-participant statistics. After TBSS for FA was completed, results were used to generate skeletonized RD and AD data for additional voxel-wise group comparisons using the TBSS non-FA pipeline (see https://fsl.fmrib.ox.ac.uk/fsl/fslwiki/TBSS/UserGuide#Using_non-FA_Images_in_TBSS).

## Statistical analyses

For contrasting case-control differences, we run voxel-wise statistics for FA, AD and RD, separately, using a nonparametric permutation-based approach (*Randomise*, implemented in FSL, 5000 permutations). Age and sex were entered as covariates. All covariates were demeaned. The cluster-forming significance threshold was set at p ≤ 0.01, after family-wise error correction for multiple comparisons using threshold-free cluster enhancement. We chose a highly conservative threshold for FA to minimize type I errors and to account for the exploratory nature of the *post hoc* analyses.

For regions identified with TBSS (FA) and TBSS non-FA (RD/AD), we extracted the mean FA, RD and AD values of each participant at the set threshold of 0.01 for further analysis and illustration purposes.

Separated linear regression models were fitted to examine whether patients' mean values of significant TBSS (FA) or significant TBSS non-FA (AD/RD) clusters, as dependent variables, were associated with duration of illness and antipsychotic medication status (coded as yes (1)/ no (0)), as independent variables The Cohen's d effect size was computed from the t-statistics for categorial variables, and using the partial correlation coefficient, r, for continuous variables [33].

If there was an association between patients' regional mean values and antipsychotic medication, follow-up correlation analyses with current and cumulative CPZ were performed using Spearman's rank correlation rho for non-normal data.

Further analysis of regional mean values and their association with clinical measures (PANSS, CGAS, and MFQ) were performed using Pearson's product moment correlation coefficient.

Statistical tests were conducted in R, version 3.5.2 (www.r-project.org).

## Results

### Demographic and clinical data

Sample demographics and clinical characteristics separated by antipsychotic medication status of EOP patients are reported in Table 1 and Table 2, respectively. EOP patients did not differ significantly from controls in general demographic variables such as age, handedness and IQ (Table 1). In line with their clinical diagnosis, EOP patients showed higher impairment of general functioning evaluated with CGAS and exhibited significantly more depressive symptoms assessed with MFQ, relative to their healthy counterparts.

Within the patient group (Table 2), patients on antipsychotic medication were significantly older than unmedicated patients. Duration of untreated psychosis (DUP) was significantly longer in the unmedicated EOP patients in comparison to the medicated EOP patients. Other demographic and clinical variables did not differ significantly between medicated and unmedicated patients.

### TBSS analyses

Voxel-wise statistical analysis of case-control differences revealed significantly lower mean FA values in the left genu of the corpus callosum, the left anterior corona radiata (ACR), and the right superior longitudinal fasciculus (SLF) in EOP patients compared to healthy controls (see Fig 1 and Table 3). There was no increase in mean FA for the opposing contrast.

Applying the TBSS pipeline to RD and AD, did not yield significant case-control differences for RD. Yet in EOP patients relative to healthy controls, AD values were significantly lower in the left ACR, right posterior limb of the internal capsule (PLIC) and right superior fronto-occipital fasciculus (SFOF, S3 Table, available online). Lower AD in the left ACR largely overlapped with the FA findings.

For descriptive purposes, extracted mean values of all scalar diffusion measures for all significant clusters stratified by group are displayed in S1 Fig (available online). TBSS case-control differences in mean FA and mean AD (patients < controls) at a cluster-forming threshold of $p \leq 0.05$ instead of 0.01 are also presented (S2 Fig, available online).

### Linear regression analyses

For clusters identified with TBSS, we performed follow-up analyses to investigate whether patients' lower FA and AD values in these brain regions were associated with duration of illness and antipsychotic treatment (S1 Table, available online). As we found no significant

**Table 1. General sample characteristics.**

| | EOP patients overall N = 22 | Healthy controls N = 33 | Statistics group-level |
|---|---|---|---|
| **Sex, female** N (%) | 15 (68.2) | 20 (60.6) | $X^2$, p = 0.775 |
| **Age at MRI** (y) | 16.71 [16.24, 17.57] | 16.19 [15.06, 17.26] | KW, p = 0.128 |
| Range | 14.53–18.25 | 12.67–18.15 | |
| **Handedness** N (%) | | | FET, p = 0.634 |
| Right | 18 (81.8) | 30 (90.9) | |
| Left | 2 (9.1) | 2 (6.1) | |
| Missing N (%) | 2 (9.1) | 1 (3.0) | |
| **Parental Education** (y) | | | |
| Mother | 15.00 [14.25, 16.00] | 16.00 [15.00, 17.00] | KW, p = 0.106 |
| Range | 11–19 | 12–22 | |
| Missing N (%) | 0 | 2 (6.1) | |
| Father | 15.00 [12.00, 16.00] | 15.50 [15.00, 17.00] | KW, p = 0.133 |
| Range | 10–23 | 11–20 | |
| Missing N (%) | 1 (4.5) | 3 (9.1) | |
| **IQ** | 103.00 [95.50, 109.50] | 102.00 [94.75, 110.75] | KW, p = 0.922 |
| Range | 83–132 | 70–116 | |
| Missing N (%) | 3 (13.63) | 1 (3) | |
| **CGAS** | 45.00 [36.25, 53.75] | 91.00 [83.00, 95.00] | KW, **p < 0.001** |
| Range | 32–59 | 75–98 | |
| **MFQ** | 28.00 [18.00, 38.00] | 5.00 [1.00, 8.25] | KW, **p < 0.001** |
| Range | 5–52 | 0–31 | |
| Missing N (%) | 1 (4.5) | 1 (3) | |
| **BMI** ($kg/m^2$) | 20.50 [18.00, 23.15] | 20.70 [18.58, 22.18] | KW, p = 0.704 |
| Range | 15.4–35.4 | 16.7–26.0 | |
| Missing N (%) | 3 (13.6) | 5 (15.2) | |
| **Cannabis use, yes** N (%) | 7 (31.8) | 1 (3.0) | FET, **p = 0.005** |
| **Alcohol use, yes** N (%) | 12 (54.5) | 15 (45.5) | $X^2$, p = 0.700 |

*non-normal distributed data in median [interquartile range], N = Number of participants, m = male, f = female, y = years, r = right, l = left, IQ = Intelligence Quotient, BMI = Body Mass Index, CGAS = Children's Global Assessment Scale, MFQ = Mood and Feelings Questionnaire, KW = Kruskal-Wallis Test, FET = Fisher's Exact Test.

patient-control differences in RD using TBSS, no follow-up analyses using mean RD were performed. Duration of illness was not significantly associated with mean FA in any of the clusters, but we found a significant negative association with mean AD in the left ACR ($d$ = -0.48, 95% CI [-1.67, 0.74], $p$ = 0.029). However, the overall regression equation of this model was not significant (p = 0.085) and the explanatory power was low ($R^2$ = 0.15), rendering this finding most likely spurious (S1 Table, available online).

Exposure to antipsychotic medication was significantly associated with mean FA values in the left ACR ($d$ = 1.37, 95% CI [0.01, 2.68], $p$ = 0.008), showing higher mean FA in medicated relative to the unmedicated patients (Fig 2). No other regions showed significant associations between antipsychotic medication and mean FA or mean AD (S1 Table, available online). To further assess the specificity of the association between antipsychotic medication and mean FA in the left ACR, we fitted an additional linear model that accounted for handedness, age, $age^2$ and sex as potential confounds (S2 Table, available online). $Age^2$ was included to account for putative non-linear effects of age on FA. The effect of antipsychotic medication on mean FA of the left ACR remained after adjusting for additional covariates ($d$ = 1.40, 95% CI [-0.04, 2.78], $p$ = 0.020). Based on visual inspection, higher mean FA in medicated patients appeared driven

**Table 2. Patient clinical characteristics stratified by antipsychotic medication status.**

| | EOP patients Off AP at scan N = 11 | EOP patients On AP at scan N = 11 | Statistics patient-level |
|---|---|---|---|
| **Sex, female** N (%) | 6 (54.5) | 9 (81.8) | FET, p = 0.361 |
| **Age Scan** (y) | 15.94 ± 1.00 | 17.44 ± 0.65 | *t-test, p < 0.001* |
| Range | 14.53–17.25 | 16.53–18.25 | |
| **Handedness, left** N (%) | 1 (10.0) | 1 (10.0) | FET, p = 1.000 |
| **BMI (kg/m$^2$)** | 19.20 [17.40, 22.90] | 20.55 [18.93, 25.43] | KW, p = 0.457 |
| Range | 15.4–28.7 | 16.3–35.4 | |
| Missing N (%) | 0 | 3 (27.3) | |
| **CGAS** | 46.00 ± 8.00 | 43.91 ± 9.38 | t-test, p = 0.580 |
| Range | 34–59 | 32–58 | |
| **MFQ** | 30.82 ± 12.60 | 27.10 ± 13.39 | t-test, p = 0.520 |
| Range | 5–52 | 8–49 | |
| Missing N (%) | 0 | 1 (9.1) | |
| **PANSS** | | | |
| positive | 19.09 ± 3.83 | 17.18 ± 3.84 | t-test, p = 0.257 |
| Range | 12–25 | 13–26 | |
| negative | 22.09 ± 7.13 | 18.27 ± 6.96 | t-test, p = 0.218 |
| Range | 9–32 | 7–32 | |
| general | 38.73 ± 8.44 | 36.27 ± 7.89 | t-test, p = 0.489 |
| Range | 28–54 | 24–52 | |
| **Age of Onset** (y) | 14.39 ± 1.92 | 14.83 ± 2.07 | t-test, p = 0.614 |
| Range | 10–16 | 12–17.6 | |
| **DUP** (w) | 36.00 [23.00, 82.00] | 12.00 [8.00, 18.00] | KW, *p = 0.005* |
| Range | 14–227 | 3–125 | |
| **DUI** (y) | 0.75 [0.66, 1.79] | 1.76 [0.64, 4.52] | KW, p = 0.375 |
| Range | 0.47–4.53 | 0.33–5.97 | |
| **Diagnosis** | | | FET, p = 1.000 |
| SCZ | 7 | 7 | |
| SCA | 1 | 0 | |
| NOS | 3 | 4 | |
| **Antipsychotics** | | | |
| Aripiprazole | | 5 | |
| Risperidone | | 3 | |
| Quetiapine | | 3 | |
| **CPZ** | | | |
| current | | 272.3 ± 140.83 | |
| Range | | 151.52–559.44 | |
| cumulative (AP-naïve, N = 9) | 0.08 ± 0.28 | 21.6 ± 19.13 | |
| Range | 0–0.92 | 1.69–59.28 | |
| **Cannabis use, yes** N (%) | 3 (27.3) | 4 (36.4) | FET, p = 1.000 |
| **Alcohol use, yes** N (%) | 4 (36.4) | 8 (72.7) | FET, p = 0.198 |

*normally distributed data as mean ± standard deviation, non-normal distributed data in median [interquartile range], AP = antipsychotics, N = number of participants, m = male, f = female, y = years, r = right, l = left, IQ = Intelligence Quotient, BMI = Body Mass Index, CGAS = Children's Global Assessment Scale, MFQ = Mood and Feelings Questionnaire, PANSS = Positive and Negative Symptom Scale, DUP = Duration of Untreated Psychosis, DUI = Duration of illness, SCZ = schizophrenia, SCA = schizoaffective, NOS = psychosis, not other specified, CPZ = chlorpromazine equivalent, KW = Kruskal-Wallis Test, FET = Fisher's Exact Test.

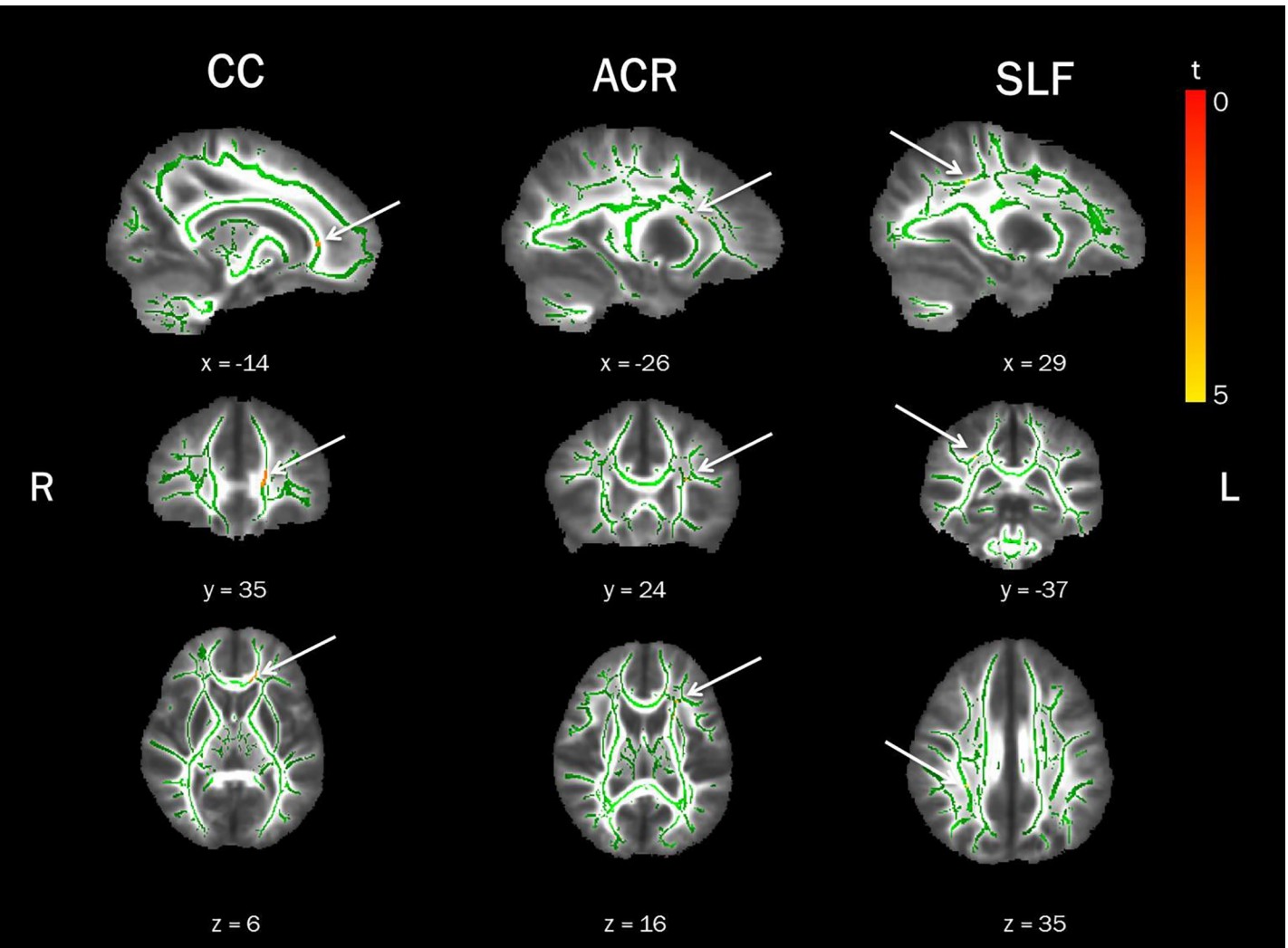

**Fig 1. Lower Fractional Anisotropy (FA) in Early Onset Psychosis (EOP) patients in comparison to healthy controls.** Displayed are significant FWE-corrected TBSS results (red-yellow, p ≤ 0.01), contrasting EOP patients against healthy controls, overlaid on the study-specific mean FA skeleton in green and the mean FA image. Results shown underwent threshold-free cluster enhancement and were corrected for age and sex. CC = corpus callosum, ACR = anterior corona radiata, SLF = superior longitudinal fasciculus, R = right, L = left.

**Table 3. White matter cluster of reduced fractional anisotropy in early onset psychosis patients relative to healthy controls.**

| CIuster | Region[1] | Side | Voxels | MNI coordinates in mm | | | t-values |
|---|---|---|---|---|---|---|---|
| | | | | X | Y | Z | |
| 4 | Genu of corpus callosum | L | 150 | -14 | 35 | 6 | 3.35 |
| 3 | Anterior corona radiata[2] | L | 46 | -26 | 13 | 14 | 3.69 |
| 2 | Superior longitudinal fasciculus | R | 12 | 29 | -37 | 35 | 4.81 |
| 1 | Anterior corona radiata (16% Inferior fronto-occipital fasciculus) [3] | L | 9 | -26 | 24 | 16 | 4.05 |

[1] Johns Hopkins University International Consortium for Brain Mapping (JHU ICBM)-DTI-81 white matter atlas and JHU white matter tractography atlas (in brackets) were utilized to label significant clusters with specific tract names.

[2] 6% uncinate fasciculus/ 5% inferior fronto-occipital fasciculus according to JHU White-Matter Tractography Atlas.

[3] 11% anterior thalamic radiation, 8% uncinate fasciculus according to JHU White-Matter Tractography Atlas.

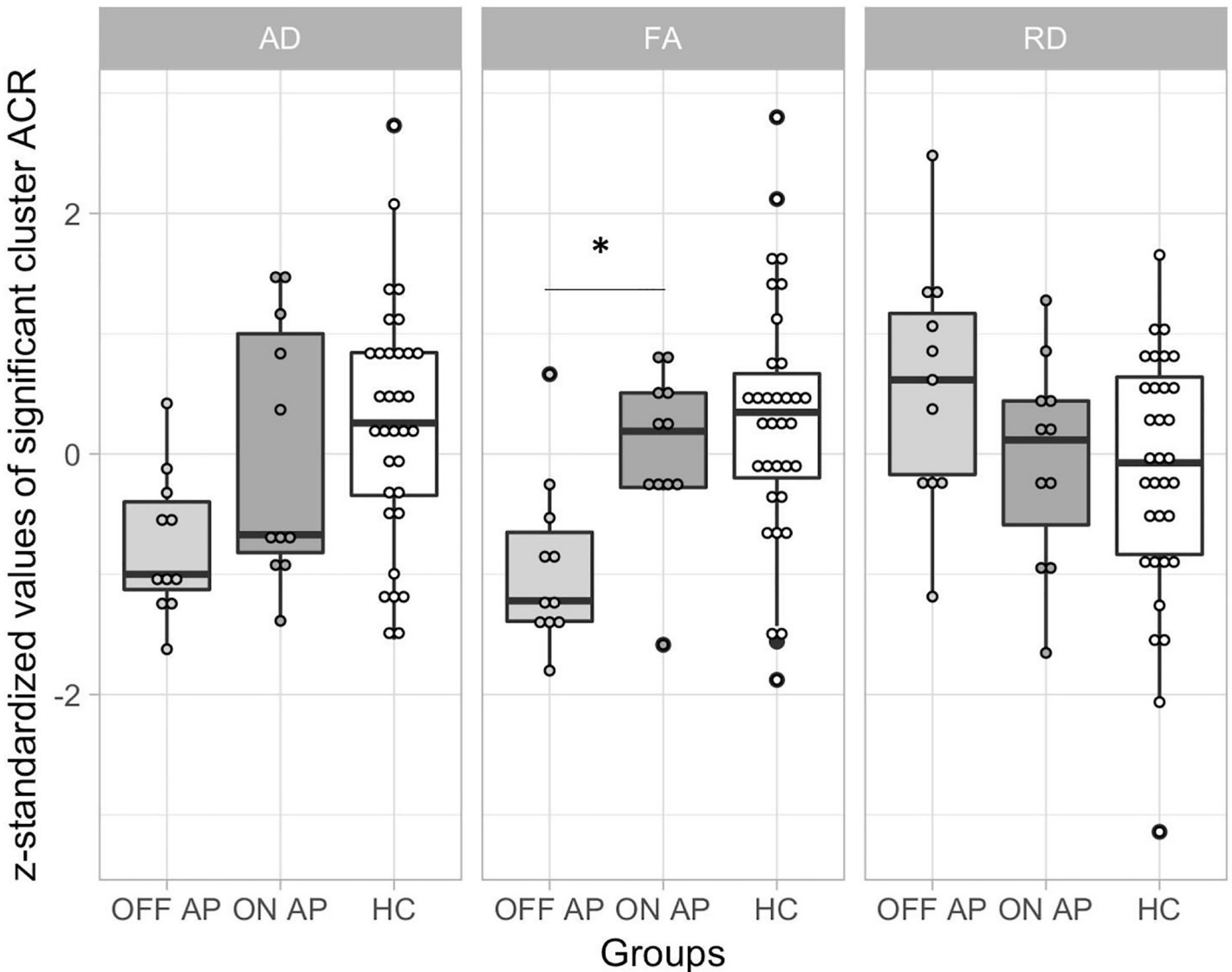

**Fig 2. Extracted scalar diffusion values of the left anterior corona radiata (ACR) cluster stratified by antipsychotic use, in comparison to the healthy controls (HC).** As scalar diffusion measures largely vary in their value ranges, the extracted mean values were z-standardized using the following formula: z = (participant's value–group mean) / standard deviation. Data is z-standardized and presented as boxplots for the different scalar diffusion measures overlaid with raw data points. HC are depicted in white, EOP patients on antipsychotic medication in dark grey and EOP patients off antipsychotic medication in light grey. EOP = Early onset psychosis, AP = Antipsychotic use (on = yes, off = no), AD = axial diffusion, FA = fractional anisotropy, RD = radial diffusion. Significant differences in scalar measures between patient subgroups, based on linear regression models, are indicated with a star.

by an increase in AD and a decrease in RD (Fig 2). However, the extracted mean AD and RD values did not differ significantly between medicated and unmedicated patients (Welch Two Sample t-test, AD: t = 0.183, p = 0.857; RD: t = 1.89, p = 0.079).

### Association with clinical measures

We found no association between current antipsychotic medication evaluated as chlorpromazine equivalent at scan day (CPZ, Spearman ρ = 0.13, n = 11, p = 0.695) or cumulative CPZ (Pearson ρ = -0.15, n = 11, p = 0.665) with regional mean FA values in the left ACR.

Furthermore, we found no significant correlations, which survived correction for multiple comparisons (FA cluster: Bonferroni, $\alpha = 0.003$; AD cluster: Bonferroni, $\alpha = 0.004$), between either extracted mean FA nor mean AD values of all significant TBSS clusters and clinical measures such as PANSS (neither positive nor negative), CGAS and MFQ scores.

## Discussion

EOP patients showed significantly lower FA and AD in largely overlapping areas, particularly in left ACR, relative to healthy controls. In patients, mean FA of the left ACR was significantly associated with antipsychotic medication status, showing higher FA values in medicated compared to unmedicated EOP patients. Indices of white matter microstructure in patients were not associated with other clinical characteristics.

Our case-control results replicate prominent brain regions with previously reported white matter abnormalities in EOP, namely corpus callosum [12], right SLF [20, 34, 35] and the left ACR [15, 16]. Lower FA in these regions appears to occur early in the disease process [16, 35, 36]. For instance, Lagopoulos and colleagues reported decreased FA in the left ACR in both patients with an established psychotic disorder and patients exhibiting prodromal symptoms, aged 14–30 years. Based on these findings, the authors proposed that abnormalities in the left ACR may be a putative precursor to the development of psychosis [16].

However, the ACR is a highly heterogeneous structure with three long-range association fiber tracts traversing through it [16]: anterior thalamic radiation (ATR), inferior fronto-occipital fasciculus (IFOF) and uncinate fasciculus (UF). All three association fibers form connections to the frontal lobe and have been implicated in the pathophysiology of psychotic disorders [16, 37–39]. In the current study, the left ACR peak voxel showed a 16% probability of IFOF involvement based on the JHU White-Matter Tractography Atlas. The IFOF connects the occipital and temporal lobes with the orbitofrontal cortex as part of the ventral visual and language stream. In particular, the left IFOF seems to subserve language semantics [40]. Already in 1996, Aloia and colleagues proposed that the disruption of semantic networks have potential implications for the origin of "thought disorder" in schizophrenia [41]. Adding to this hypothesis, patients with 22q11.2 deletion syndrome, who are genetically at high risk for developing schizophrenia, showed lower FA values in left IFOF [42]. Furthermore, DeRosse and colleagues found that lower FA proximal to the bilateral SLF and corticospinal tract, and left IFOF and left inferior longitudinal fasciculus (ILF), were associated with higher levels of psychotic-like experiences in otherwise healthy volunteers [43]. In early-onset schizophrenia (EOS) patients, lower FA in the left IFOF and the left ILF were associated with worse neurocognitive performance [15]. The authors also detected a shared decrease in FA in the left IFOF among patients with clinical high risk for schizophrenia and patients with established EOS, in comparison to healthy controls. Taken together, these findings suggest that white matter abnormalities in the left ACR, putatively in the left IFOF, may represent a potential candidate for understanding the etiology of psychosis.

This assumption seems further supported by effects of antipsychotic medication on diffusion metrics in the left ACR. We found that FA values in the left ACR were significantly associated with antipsychotic medication status, showing higher FA values in medicated relative to unmedicated EOP patients. No such association was found with the other brain regions showing significantly lower FA values. Besides the high Cohen's d effect size estimate, we found no significant association of regional FA with either current or cumulative antipsychotic exposure. These null-findings in line with previous studies in EOP patients [12, 18–21] and studies in adult-onset psychosis [5, 22]. For instance, studying white matter microstructure in 1,963 patients with schizophrenia, Kelly and colleagues found no significant effect of medication

dosage [5]. Hence, the usage of antipsychotic medication rather than the actual dose might induce the observed changes in white matter microstructure.

In medicated patients, increased FA values appears driven by an increase in AD and a decrease in RD, relative to their unmedicated counterparts (Fig 2). Thus, FA might be enhanced by antipsychotic medication as a result of both facilitated parallel diffusion (AD, potentially mediated by an increase in axon numbers, and restricted perpendicular diffusion (RD), indicative of changes in myelin.

Converging evidence from multiple studies suggests oligodendroglial dysfunction, with subsequent abnormalities in myelin maintenance and repair, to underpin white matter abnormalities observed in psychotic patients [44]. In the framework of schizophrenia, it has been proposed that myelin dysfunction, especially in frontal regions, contributes to psychotic symptoms [19, 44]. Based on findings from cell culture studies using aripiprazole [45] and rodent work using quetiapine [46, 47], second-generation antipsychotic medication may promote oligodendrocyte recovery and myelin repair, which could lead to improved white matter integrity and, subsequently, reduced psychotic symptom load. A recent MRI study in patients with schizophrenia also reports on promyelinating effects of antipsychotics [48]. Tishler and colleagues found an increase in intracortical myelin predominantly mediated by risperidone and other second-generation antipsychotics in adult patients with schizophrenia compared to healthy controls within the first year of treatment. In the current study, medicated EOP patients were treated with either received aripiprazole, quetiapine or risperidone. One might speculate that use of second-generation antipsychotics early in the disease process might affect white matter microstructure by remediating oligodendroglial dysfunction, reflected by an increase in FA detected by DWI. However, given the small sample size of the current study, no firm conclusions can be drawn.

Even though FA is highly sensitive to microstructural changes in general, it lacks neurobiological specificity to the exact type of change [6]. For instance, a decrease in FA can reflect alternations in fiber organization, including packing density and fiber crossing, and myelin loss or myelin remodeling [49]. The interpretation of FA and its scalar sub-measures is further complicated by the lack of sensitivity in regions of crossing white matter tracts, such as the ACR. Yet, as stated by Alexander and colleagues [6], this confound is unavoidable as many areas of the brain have considerable areas of fiber crossing. Here, we found a widespread decrease of AD on a whole brain level, indicative of axonal damage, but there were no changes in RD, relative to healthy controls. These findings are not in line with our a priori hypothesis of increased RD and unchanged AD, and previous work by Lagopoulos and colleagues, reporting decreased FA in the left ACR driven by increases in RD [16].

Although there are likely several reasons for these conflicting findings, neuroinflammatory processes in psychosis might impede the interpretation of white matter microstructural changes in EOP patients relative to healthy controls [50]. For instance, in an animal model of cuprizone-induced demyelination of corpus callosum, regions with extensive axonal edema and prominent cellular inflammation showed no change in RD, while AD values were diminished at the beginning of demyelination [51]. Given the neuroinflammation hypothesis of schizophrenia [52], it seems likely that the disease progression encompasses a dynamic evolution of inflammation, axonal injury, and myelin degeneration. In the current study, one might speculate that the brain of EOP patients is on the verge of undergoing demyelination processes, reflected by widespread decreases in AD. However, the timing of neuroinflammation in psychotic disorders relative to tissue injury is unclear, leading to a heightened risk of misinterpreting changes in DWI measures. According to a recent review, DWI seems to underestimate the extent of demyelination (undervalued RD), and overestimate the extent of axonal injury (overvalued AD), when neuroinflammation is linked to tissue damage [53]. This pattern seems

replicated in our study, with significant changes in AD and no changes in RD. As the consistency of DWI metrics seems affected by brain edema and inflammatory response, future studies can benefit from using tools such as free water imaging. Free-water imaging separates the contribution of extracellular water from the diffusion measure, leading to a higher specificity in detecting microstructural changes [22].

Deviations in scalar DWI measures in the current study compared to previous studies could also be due to ongoing white matter maturation processes in our adolescent EOP sample. In healthy individuals, age-related increases in FA during childhood, adolescence and early adulthood have been consistently reported [9, 10, 54, 55]. This increase in FA seems primarily driven by a reduction in RD, while AD remains fairly stable or decreases slightly [56, 57]. Findings for AD changes during the transition to adulthood are less consistent [9, 10, 54, 55]. Thus, the AD difference found in the current study could also be attributed to developmental processes, which may fade as adolescents mature into adulthood.

However, neurodevelopmental trajectories of white matter structure relative to disease progression in EOP patients are unclear. So far, three different studies yielded inconclusive results, either postulating diverging [19], converging [58], or parallel [59] trajectories relative to healthy controls. In the current study, we did not find any association between duration of illness and regional FA. This is in line with previous findings from Kumra and colleagues, who speculated that lower FA in EOP patients compared to healthy controls reflects developmental abnormalities rather than secondary effects of the disease progression [19]. In addition, Epstein and Kumra found lower FA in the inferior longitudinal fasciculus, IFOF and corticospinal tract, but no significant group differences in longitudinal changes in FA [59]. Thus, the observed changes in the current study might persist through adolescence and adulthood, but do not affect the overall white matter maturation trajectories.

The results of the current study should be considered in the context of several limitations. The cross-sectional nature of the current study precludes the assessment of developmental effects over time. The small sample size implies that the results should be interpreted with caution, and that findings require replication in a larger sample. While symptom severity did not differ between medicated and unmedicated patients at time of scan, prior symptom severity likely influenced the initiation of antipsychotic treatment, which may have influenced our results. Unmedicated EOP patients were significantly younger than those receiving medication, and although the analyses were corrected for age, time-of-measurement effects, with older patients having higher FA values than younger patients due to more advanced white matter maturation, may have confounded our results. However, we did not find significant differences in mean FA values between medicated and unmedicated patients in other brain regions showing a similar maturation trajectory than the ACR, such as the SLF [56]. DWI data were acquired at an in-plane resolution of 1.875 x 1.875 mm which could result in less precise estimate of the diffusion tensor and the subsequent scalar diffusion measures [60]. We acknowledge that the rather large anisotropic voxel size in the current study may have influenced the results, and that higher resolution DWI should be used for future studies.

Strengths of this study include a well-characterized and balanced sample of medicated and unmedicated EOP patients and age-matched healthy controls, as well as standardized diffusion MRI acquisition and robust statistical analyses approaches. While we acknowledge the possibility that our results might be spurious and limited to our EOP sample, we consider this unlikely to be the driving factor for the following reasons: (i) we replicated white matter microstructure abnormalities in brain regions implicated in the pathophysiology of EOP [17] and adult-onset schizophrenia [5] using a stringent significance threshold of 0.01, and (ii) medicated EOP patients received either aripiprazole, quetiapine or risperidone, drugs which have

previously been associated with changes in white matter microstructure in cell-culture [45], rodent [46, 47] and human studies [48].

In summary, the present study shows lower FA and AD in patients relative to controls and is the first to link antipsychotic medication status to altered regional FA in the left ACR in patients with EOP. Understanding the significance of white matter abnormalities in the left ACR in adolescents with EOP and the putative effect of antipsychotic medication, may help to phenotype the disease and to improve treatment regimes. Building on our tentative results, longitudinal studies with larger samples sizes using high resolution DWI in combination with clinical, genetic and neurocognitive measures are warranted to delineate heritability, affected brain regions, antipsychotic medication effects, and white matter microstructural differences over time.

## Supporting information

**S1 Fig.** Extracted mean fractional anisotropy (FA), axial diffusivity (AD) and radial diffusivity (RD) values of significant FA (A) and AD clusters (B) identified with whole-brain TBSS. Data is presented as grey boxplots for early onset psychosis (EOP) patients and white boxplots for healthy controls (HC). ACR = anterior corona radiata, CC = corpus callosum, SLF = superior longitudinal fasciculus, PLIC = Posterior limb of the internal capsule, SFOF = superior fronto-occipital fasciculus. Note: Data is presented for descriptive purpose only.
(DOCX)

**S2 Fig. Lower Fractional Anisotropy (FA) and Axial Diffusivity (AD) in early onset psychosis (EOP) patients in comparison to healthy controls.** Displayed are significant FWE-corrected TBSS results for FA (light blue–dark blue, $p \leq 0.05$) and AD (red-yellow, $p \leq 0.05$), contrasting EOP patients against healthy controls, overlaid on the study-specific mean FA image. The overlap of both DWI measures is depicted in green. Results shown underwent threshold-free cluster enhancement and are corrected for age and sex. CC = corpus callosum, ACR = anterior corona radiata, SLF = superior longitudinal fasciculus, R = right, L = left. Note: Data is presented for descriptive purpose only.
(DOCX)

**S1 Table. Results of linear regression models with extracted mean Fractional Anisotropy (FA) and mean Axial Diffusivity (AD) values of significant TBSS clusters.**
(DOCX)

**S2 Table. Extended linear regression model for extracted mean Fractional Anisotropy (FA) of the left anterior corona radiata.**
(DOCX)

**S3 Table. White matter cluster of reduced axial anisotropy in early onset psychosis patients relative to healthy controls.**
(DOCX)

## Acknowledgments

We thank the study participants and the Youth-TOP clinicians involved in recruitment and assessment at the Norwegian Centre for Mental Disorders (NORMENT) and the Diakonhjemmet Hospital, Oslo, Norway (Runar Elle Smelror, Kirsten Wedervang-Resell, Cecilie Haggag Johannessen, Tarje Tinderholt, Tove Matzen Drachmann). Further, we like to thank Kristine Engen and Brian Frank O'Donnell for proofreading.

## Author Contributions

**Conceptualization:** Claudia Barth.

**Data curation:** Claudia Barth, Vera Lonning.

**Formal analysis:** Claudia Barth, Tiril Pedersen Gurholt.

**Funding acquisition:** Ole A. Andreassen, Anne M. Myhre, Ingrid Agartz.

**Investigation:** Claudia Barth.

**Methodology:** Claudia Barth, Tiril Pedersen Gurholt.

**Project administration:** Vera Lonning, Anne M. Myhre.

**Resources:** Ole A. Andreassen, Anne M. Myhre, Ingrid Agartz.

**Software:** Claudia Barth.

**Supervision:** Ingrid Agartz.

**Visualization:** Claudia Barth.

**Writing – original draft:** Claudia Barth.

**Writing – review & editing:** Claudia Barth, Vera Lonning, Tiril Pedersen Gurholt, Ole A. Andreassen, Anne M. Myhre, Ingrid Agartz.

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
