## [Decision Letter · Decision Letter 0]

3 Mar 2020

Pécs, Hungary

March 2, 2020

PONE-D-20-03059

Impact of second-generation antipsychotics on white matter microstructure in adolescent-onset psychosis.

PLOS ONE

Dear Dr. Barth,

Thank you for submitting your manuscript to PLOS ONE. After careful consideration, we feel that it has merit but does not fully meet PLOS ONE’s publication criteria as it currently stands. Therefore, we invite you to submit a revised version of the manuscript that addresses the points raised by the Reviewers, listed below.

We would appreciate receiving your revised manuscript by Apr 17 2020 11:59PM. To enhance the reproducibility of your results, we recommend that if applicable you deposit your laboratory protocols in protocols.io, where a protocol can be assigned its own identifier (DOI) such that it can be cited independently in the future. For instructions see: http://journals.plos.org/plosone/s/submission-guidelines#loc-laboratory-protocols

We look forward to receiving your revised manuscript.

Kind regards,

Joseph Najbauer, Ph.D.

Academic Editor

PLOS ONE

Journal Requirements:

Reviewers' comments:

Reviewer's Responses to Questions

**Comments to the Author**

1. Is the manuscript technically sound, and do the data support the conclusions?

Reviewer #1: Yes

Reviewer #2: Partly

2. Has the statistical analysis been performed appropriately and rigorously? 

Reviewer #1: Yes

Reviewer #2: Yes

3. Have the authors made all data underlying the findings in their manuscript fully available?

Reviewer #1: No

Reviewer #2: Yes

4. Is the manuscript presented in an intelligible fashion and written in standard English?

Reviewer #1: Yes

Reviewer #2: Yes

5. Review Comments to the Author

Reviewer #1: Review PONE-D-20-03059

Overall this manuscript is well written, rigorous and in a logical order. There are several remarks that need to be addressed. These are provided per section and require changes to be made in order to make the manuscript acceptable.

Introduction

The introduction is logical in understanding, but should start with a statement of why this research is important. Please provide numbers on why there is a need to do research in psychotic disorder.

In the introduction the FA measure is introduced. This might measure some “integrity” of white matter. Introduce why it is needed to investigate RD and AD. Next explain shortly what is known about RD and AD in psychotic disorder. Generally RD is increased in patients with psychotic disorder, while AD is unchanged compared to healthy controls. This leads to the hypotheses stated in the last part of the introduction.

It needs to be noted that there is critical developmental phase and several factors play a role in the aberrant development. Note that brain white matter develops till the age of 25 and that there is a difference between males and females. This might be related to a more frequent early onset psychosis in males.

Methods and materials

A total of 27 patients and 40 controls were included which underwent MRI examination. Why is the control group larger compared to the patient group? Based on what information was this sample size determined?

It was clearly described why participants were excluded in several steps in the study. What are strong motion artefacts in the DWI data? How was this quantified?

In the MRI data acquisition section it is stated that “The diffusion imaging data was...”. The word data is plural and it should therefor read “The diffusion imaging data were...”. The MRI data are acquired with an anisotropic voxel size (2.5x1.875,1.875) which could have lead to a less precise estimate of the diffusion tensor. This is an issue that needs to be discussed.

The diffusion data analysis was done in a robust and fairly standard way.

In the manuscript the word gender was used, while the word sex more precisely describes the biological meaning of it. This needs to be replaced.

In the analyses handedness was not taken into account as a covariate. Since the results report on left/right differences handedness needs to be added as a covariate.

Significant findings from the TBSS analyses were extracted and further analysed by extracting mean FA, AD and RD. These values were associated with illness duration. Why was this analyses conducted separately? This can also be done within TBSS and only in the patient group. Is it expected that there first is a group difference and next an association with duration on this difference between groups? It seems more logical to consider the illness duration within the patient group. Additionally, the illness duration might not be normally distributed within the group and needs to be reported (normal distribution test).

Clinical measures were associated with DWI measures within the patient group. This sample of 22 patients is very limited to test three DWI measures associated with three clinical measures. There are multiple tests in a small group and there is a risk of falsely reporting positive findings. Since a proper sample size calculation is missing, the results of this analyses should be interpreted with caution.

Results

Please share the statistical output from the TBSS analyses. Currently the manuscript has 2D images and the reader can be informed fully by providing the 3D images in MNI space. This allows for statistical pooling with other cohorts and meta-analyses. For example this can be shared on FigShare. Note that these data are anonymous, because only statistical maps are shared.

A description in text is missing for tables 1 and 2. Add a few short sentences on the sample characteristics and patient samples.

In the results about the TBSS analyses it is reported that there are different mean values per group. It is unclear if these are mean values per group or that these values were not reported. If possible, report the mean values in text.

The comparison between medicated and unmedicated EOP patients is fairly week due to the fact that both groups only have 11 individuals. Is there enough power to detect small differences in DWI measures?

Discussion

First start with a short recap of the main findings and next add previous findings.

In line 311 it is stated that a prediction was made. With the available data prediction is impossible. This needs to be rephrased. In line 315 and 316 a “This lack of significant associations” was described. This needs to be rephrased as as “This null-finding”. It is important to report null-findings a bit more positive.

The statement that medication leads to an increase in FA detected by DWI is an over-interpretation of the results. First, there is no causality in the findings and therefor the word “leading” in line 339 needs to be replaced. Furthermore, a sharp note needs to be made on the small sample size in this part of the discussion. Readers might interpret this part of the discussion without taking note of the limited sample size behind it. This is at maximum an exploratory part of the study.

The sample size limitation is shortly discussed, without mentioning anything about an independent sample size calculation that would be needed. As this is post-hoc, nothing can be done about the results and using a p-value threshold at < 0.01 was already required for the multiple tests (for FA, AD and RD) that were preformed.

The conclusion is logically following based on the methods and results presented in the manuscript.

Reviewer #2: In this study the authors used diffusion weighted imaging to examine white matter structure in people with early onset psychosis, and the effects of current antipsychotic use on WM. In this small sample, they found reduced FA in the EOP group, and WM differences in people on vs. off antipsychotic medications at testing, with the medicated group showing higher FA than the unmedicated group. Ascertaining EOP participants on and off medications is challenging, and investigation of both EOP generally and medication effects on WM specifically are both valuable. Several methodological notes are offered.

1. This is obviously a very small sample, especially when dividing the EOP group into medicated and unmedicated subsamples. Effect sizes and confidence intervals should be reported for all main analyses, and findings should be interpreted as tentative.

2. The medication-related analyses were exploratory with no ad-hoc hypotheses; this obviously has implications for interpretation, and (again) conclusions should be tentative. Oddly, in the discussion the authors imply that the findings make sense based on the literature, so it is puzzling that they do not make a priori hypotheses based on this literature.

3. The title focus seems a bit odd, as the stated aims in the manuscript were to 1) look at WM scalar measures in EOP and then 2) exploratory analysis of medication effects. The title (which makes medication effects central) should reflect the actual aims of the paper.

4. While the FA findings were hypothesized, the authors found no difference in RD (contrary to hypothesis), and decrease AD (not hypothesized). Again, in a very small sample it should be noted that these were not expected findings, and await replication.

5. Though not statistically significant (likely due to small sample size), the off medication group had higher symptom severity than the on medication group. How was clinical severity controlled in the analyses?

6. Increased FA was reported in medicated compared to unmedicated patients; however, no association with CPZ load was found (which is a bit confusing). The authors attempt to address this in the discussion, but I think some degree of caution is needed here.

7. The z-standardized values of AD in the patient groups (but not controls) are very highly skewed, as is the “off antipsychotic” group FA data. What statistical procedures were included to account for this issue?

8. Some minor English language editing would be useful.

6. PLOS authors have the option to publish the peer review history of their article (what does this mean?). If published, this will include your full peer review and any attached files.

Reviewer #1: Yes: Stijn Michielse, PhD

Reviewer #2: No

---

## [Author Response · Author response to Decision Letter 0]

17 Mar 2020

See uploaded file titled "ResponsesToReviewer.docx" at the bottom of the merged PDF file.

---

## [Decision Letter · Decision Letter 1]

9 Apr 2020

Pécs, Hungary

April, 9,  2020

PONE-D-20-03059R1

Exploring  white matter microstructure and the impact of antipsychotics in adolescent-onset psychosis.

PLOS ONE

Dear Dr. Barth,

Thank you for submitting your manuscript (R1 version) to PLOS ONE. After careful consideration, we feel that it has merit but does not fully meet PLOS ONE’s publication criteria as it currently stands. Therefore, we invite you to submit a revised version of the manuscript that addresses the points raised by Reviewer #1, listed below.

We would appreciate receiving your revised manuscript by May 24 2020 11:59PM. To enhance the reproducibility of your results, we recommend that if applicable you deposit your laboratory protocols in protocols.io, where a protocol can be assigned its own identifier (DOI) such that it can be cited independently in the future. For instructions see: http://journals.plos.org/plosone/s/submission-guidelines#loc-laboratory-protocols

We look forward to receiving your revised manuscript.

Kind regards,

Joseph Najbauer, Ph.D.

Academic Editor

PLOS ONE

Reviewers' comments:

Reviewer's Responses to Questions

**Comments to the Author**

1. If the authors have adequately addressed your comments raised in a previous round of review and you feel that this manuscript is now acceptable for publication, you may indicate that here to bypass the “Comments to the Author” section, enter your conflict of interest statement in the “Confidential to Editor” section, and submit your "Accept" recommendation.

Reviewer #1: (No Response)

Reviewer #2: All comments have been addressed

2. Is the manuscript technically sound, and do the data support the conclusions?

Reviewer #1: Yes

Reviewer #2: Yes

3. Has the statistical analysis been performed appropriately and rigorously? 

Reviewer #1: Yes

Reviewer #2: Yes

4. Have the authors made all data underlying the findings in their manuscript fully available?

Reviewer #1: No

Reviewer #2: Yes

5. Is the manuscript presented in an intelligible fashion and written in standard English?

Reviewer #1: Yes

Reviewer #2: Yes

6. Review Comments to the Author

Reviewer #1: Review PONE-D-20-03059 R1

The manuscript has greatly improved with the implementation of the changes. Most of the comments were correctly realized. A few items remain;

- In the methods section indeed it was clarified that 67 participants were recruited within the context of a larger longitudinal study. It still remains unclear why the healthy controls were oversampled compared to the patients. The comparison 27 patients versus 40 controls needs to be clarified. Was the expected effect size in patients almost twice the amount as in the controls? Why was this not an equal comparison of 27 patients with 27 controls? Also, perhaps the controls could have been matched with the patients based on age, sex and handedness.

- The point about the anisotropic voxel size was partially addressed. There are limitations to the tensor fitting model and an anisotropic voxel size can influence the outcome measures. This point needs to be explained as a limitation to the current study. See DOI: 10.1016/j.neuroimage.2012.06.081 for more detail on this topic.

- It was decided not to include handedness as a covariate in the analysis. The authors agree that handedness may influence the bilateral results. Indeed there were no significant group differences on this variable, which is reassuring. Though, when assessing more detail in the comparison between the on medication and off medication patient groups it might be an issue. Most patients were right-handed which implies that the white matter in the right hemisphere is more developed. Therefore, findings might point to changes in the associations with white matter tracts in the right hemisphere. This could go either direction as there might be an interaction with medication use. In order to get an indication of how handedness might influence the findings, it is encouraged to include a sensitivity analysis. This can be done by extending the linear regression analyses on the TBSS clusters. It is fine to add this to the supplementary materials and shortly mention it in the results. In this way the main analyses can remain as it is and the sensitivity analysis can be added to inform about the influence of handedness on the results.

- The comment on the sample size is addressed by stating that there is a lack of previous studies, and indeed there are limitations in this research field. This contradicts with the introduction part; “So far, studies in EOP patients do not indicate an impact of either current (18-20) or cumulative antipsychotic exposure (12, 19, 21) on scalar DWI measures. “. The studies in references 12, 18-21 do report on the association between antipsychotic medication use and DWI measures. To at least have an indication of the expected effect, an estimated effect size should be calculated based on the available data.

- It is reassuring to read that there is an intention to share the data via NeuroVault. Would it be possible to have the link to NeuroVault integrated in the manuscript, before publication? Otherwise readers still need to search for the link after publication.

Reviewer #2: (No Response)

7. PLOS authors have the option to publish the peer review history of their article (what does this mean?). If published, this will include your full peer review and any attached files.

Reviewer #1: Yes: Stijn Michielse

Reviewer #2: No

---

## [Author Response · Author response to Decision Letter 1]

17 Apr 2020

See uploaded Response to Reviewer file.

---

## [Decision Letter · Decision Letter 2]

12 May 2020

Pécs, Hungary

May 10, 2020

Exploring  white matter microstructure and the impact of antipsychotics in adolescent-onset psychosis.

PONE-D-20-03059R2

Dear Dr. Barth,

We are pleased to inform you that your manuscript (R2 version) has been judged scientifically suitable for publication and will be formally accepted for publication once it complies with all outstanding technical requirements.

With kind regards,

Joseph Najbauer, Ph.D.

Academic Editor

PLOS ONE

Reviewers' comments:

Reviewer's Responses to Questions

**Comments to the Author**

1. If the authors have adequately addressed your comments raised in a previous round of review and you feel that this manuscript is now acceptable for publication, you may indicate that here to bypass the “Comments to the Author” section, enter your conflict of interest statement in the “Confidential to Editor” section, and submit your "Accept" recommendation.

Reviewer #1: All comments have been addressed

2. Is the manuscript technically sound, and do the data support the conclusions?

Reviewer #1: Yes

3. Has the statistical analysis been performed appropriately and rigorously? 

Reviewer #1: Yes

4. Have the authors made all data underlying the findings in their manuscript fully available?

Reviewer #1: No

5. Is the manuscript presented in an intelligible fashion and written in standard English?

Reviewer #1: Yes

6. Review Comments to the Author

Reviewer #1: Thanks for properly addressing the additional comments on the manuscript. It greatly improved, and the data shall be shared openly.

7. PLOS authors have the option to publish the peer review history of their article (what does this mean?). If published, this will include your full peer review and any attached files.

Reviewer #1: Yes: Stijn Michielse

---

## [Editor Report · Acceptance letter]

21 May 2020

PONE-D-20-03059R2 

Exploring white matter microstructure and the impact of antipsychotics in adolescent-onset psychosis. 

Dear Dr. Barth:

I am pleased to inform you that your manuscript has been deemed suitable for publication in PLOS ONE. Congratulations! Your manuscript is now with our production department. 

With kind regards,

on behalf of

Dr. Joseph Najbauer 

Academic Editor

PLOS ONE